# HLAIIPred: cross-attention mechanism for modeling the interaction of HLA class II molecules with peptides

Mojtaba Haghighatlari ®[1] ✉, Nicholas Marze[2], Robert Seward[3], Andrew Ciarla[3], Rachel Hindin[3], Jennifer Calderini[3], Benjamin Keenan[3], Santosh Dhule[3], Sarah Hall-Swan[1], Timothy P. Hickling ®[2], Eric Bennett ®[2], Brajesh Rai ®[1] & Sophie Tourdot[3]

We introduce HLAIIPred, a deep learning model to predict peptides presented by class II human leukocyte antigens (HLAII) on the surface of antigen presenting cells. HLAIIPred is trained using a Transformer-based neural network and a dataset comprising of HLAII-presented peptides identified by mass spectrometry. In addition to predicting peptide presentation, the model can also provide important insights into peptide-HLAII interactions by identifying core peptide residues that form such interactions. We evaluate the performance of HLAIIPred on three different tasks, peptide presentation in monoallelic samples, immunogenicity prediction of therapeutic antibodies, and neoantigen prioritization for cancer immunotherapy. Additionally, we created a dataset of biotherapeutics HLAII peptides presented by human dendritic cells. This data is used to develop screening strategies to predict the unwanted immunogenic segments of therapeutic antibodies by HLAII presentation models. HLAIIPred demonstrates superior or equivalent performance when compared to the latest models across all evaluated benchmark datasets. We achieve a 16% increase in prediction of presented peptides compared to the second-best model on a set of unseen peptides presented by less frequent alleles. The model improves clinical immunogenicity prediction, identifies epitopes in therapeutic antibodies and prioritize neoantigens with high accuracy.

Major histocompatibility complex (MHC) consists of a group of genes that code for proteins involved in the presentation of antigens to T cells. In humans, the MHC is also known as the human leukocyte antigen (HLA), and is divided into two classes, HLA class I (HLAI) and HLA class II (HLAII). HLAII molecules present peptides derived from extracellular proteins to CD4+ T cells. Our adaptive immune system relies substantially on the CD4+ T cell responses to eliminate invasive components of pathogens or cancer cells[1–5]. The peptides that are presented by HLAII are generated by the proteolytic degradation of extracellular proteins, and they are typically 8-26 amino acids (AAs) in length. These peptides are then loaded onto HLAII, and the peptide-HLAII complex (pHLAII) is transported to the surface of antigen presenting cells, where it can be recognized by CD4+ T cells[6]. From a genomic perspective, there are three main classes of HLAII: the HLA-DR, HLA-DP, and HLA-DQ gene loci. Each gene locus encodes thousands of alpha and beta chains that will be combined to form heterodimer proteins with diverse peptide presentation specificities, which

is essential for the immune system to recognize a wide range of antigens. Throughout the text we refer to these locus-specific heterodimers as alleles (i.e., DR, DP, and DQ alleles).

Accurate characterization of peptides that are likely to be presented by HLAII is necessary to derisk immunogenicity associated with therapeutic proteins (e.g. mAbs) or to modulate immunogenicity of vaccine design. However, experimental methods for measuring peptide-HLAII binding or their stable presentation are time-consuming and expensive, making them unsuitable for high-throughput screening of large numbers of biologics or vaccine candidates in early discovery stages. While machine learning methods for predicting peptide-HLAII binding have been developed to address this problem[7–14], developing predictive models for peptide presentation by HLA-II molecules remains a challenging task for the following reasons:

- **Lack of negative samples:** While experimental measurements such as those based on mass spectrometry (MS) provide a rich dataset of

---

[1]Machine Learning and Computational Sciences, Pfizer Research and Development, Cambridge, MA, USA. [2]Biomedicine Design, Pfizer Research and Development, Cambridge, MA, USA. [3]Pharmacokinetics, Dynamics and Metabolism, Pfizer Research and Development, Andover, MA, USA. ✉e-mail: mojtaba.haghighatlari@pfizer.com

HLAII-presented peptides, a similar collection of peptides that are known to be not presented by HLAII is not available, making the task of training machine learning based classification models particularly challenging.

- **Polyallelic nature of the data:** Available experimental pHLAII data is usually polyallelic, which means that one or more of the existing alleles in a patient or donor sample (up to 14 alleles) can be potentially responsible for the peptide presentation. Although polyallelic data increases the diversity of HLAII alleles in response to antigens, it complicates the training of machine learning models due to the non-deterministic mapping between alleles and presented peptides in training data. Recently, the number of monoallelic data has increased[15–17], which resolves this issue to some extent.

- **Peptide binding core prediction:** Although the length of peptides presented by HLAII can vary from 8 to 26 AAs, the majority of the binding occurs in a contiguous segment of 9 AAs, which is referred to as the peptide binding core[18]. Accurate prediction of these residues is a challenging problem due to the lack of such data for most peptides in training or evaluation datasets (only a small number of peptides have been structurally characterized[19]). We seek to identify binding cores along with peptide presentation in order to: i) incorporate prior biological knowledge in the model training stages, ii) enable the alignment of peptides by core residues to identify likely binders at each position, iii) increase model specificity for screening full-length proteins, and iv) improve model interpretability.

Prior studies have addressed these challenges using different design methodologies and machine learning models. A common approach adopted by several studies is to construct negative examples using random decoys, which are peptide segments derived from human proteome[11,14,16]. Alternatively, one can use one-class classification[20–22] that train the model only using positive samples. The former approach is prone to leak the information from positive samples into the negative class, while the latter approach usually lacks the generalizability.

Unambiguous peptide-allele association, due to the polyallelic nature of the dataset, introduces another complexity in developing a HLAII-peptide presentation model (Challenge 2). Previous publications have attempted to overcome this challenge by either breaking the polyallelic data into monoallelic data based on a prior knowledge[23] or by using probabilistic motif deconvolution or pseudo-labeling[13,24]. Whereas, other studies have used the polyallelic data as-is and relied on the model to learn the peptide-allele association based on the highest score among all alleles[25]. More recent studies have used the monoallelic data to train the model and then fine-tune it on the polyallelic data to improve the generalization of the model[26,27].

Regarding the third challenge, usually the binding cores are not explicitly predicted and the core is selected based on the highest score among all 9mer cores of the peptide[16,25]. More recently, an AlphaFold-based pipeline has shown promising performance in predicting the core residues of peptides that interact with HLAII molecules[19]. In terms of model architecture, studies have used different types of models, including recurrent neural networks (e.g., LSTM[28]), Transformers[29], graph neural networks, and convolutional neural networks[14,30,31]. All these models use the peptide AA sequence as input, however the graph-based models also take advantage of predefined intermolecular connections between AA residues.

In this work, we introduce HLAIIPred, a deep learning model for the prediction of pHLAII presentation. HLAIIPred is based on the cross-attention mechanism and Transformer model architecture, which have been shown to be effective for a wide range of tasks that process sequential inputs[29]. The model architecture is customized to not only infuse biological constraints, but flexibly learn pairwise interactions between peptides and HLAII molecules. We leveraged data preprocessing and regularization techniques to reduce the bias towards dominant binding cores and over-represented HLAII alleles. The training data is collected from public databases with HLAII peptidomics data from MS experiments.

## Results
### Immunopeptidomics datasets are highly imbalanced

As described in the Methods section, the peptidomics data used in this study was obtained from multiple sources, comprising 597508 unique peptides from 341 samples with 172 unique alleles, including 54 monoallelic samples. The distribution of data based on the source, gene loci, and allele is shown in Fig. 1.

For data splitting, we need to deal with paired inputs (peptide sequences and HLAII alleles) that are interdependent and highly imbalanced. Any bias towards dominant binding cores or over-represented HLAII alleles reduce the generalizability of the model. We used the Thrift et al. splitting strategy[25] and aimed to minimize 9mer overlap and balance gene ontology terms in each subset simultaneously. Although this approach is an important step towards a fair and systematic data splitting, it fails to address the data imbalance resulting from peptide similarity within each split. Clustering of the data based on the 80% peptide sequence similarity (see Methods for details of the clustering approach) shows that only 8.1% of the peptides appear in a cluster of size 1 (Fig. 1C). A similar percentage of the data (7.9%) are in clusters with size greater than 500. These numbers clearly confirm the skewed distribution of data in favor of similar peptides and dominant binding cores. To address this issue, we assigned a weight to each data point based on the inverse size of its respective cluster. Since identical peptides may be presented by different alleles in different samples, the total cluster size is determined by the number of each cluster ID in all peptide-allele(s) pairs. Ultimately, these weights are used to modify the loss function with the goal of improving generalization accuracy of our model despite such imbalance in the training set.

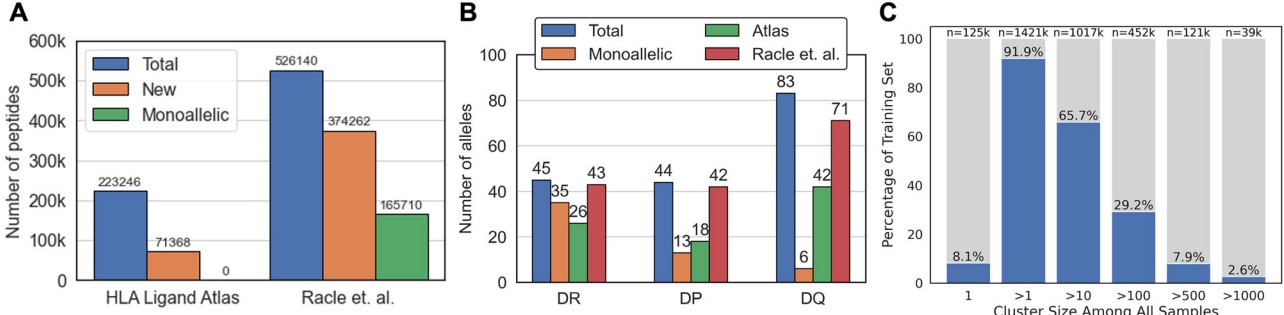

**Fig. 1 | Overview of the Immunopeptidomics datasets. A** Distribution of unique peptides by the source of data. "New" data are those that are not included in the other source of data. Monoallelic data are only compiled and reported by Racle et al. dataset. **B** Distribution of unique HLAII alleles grouped by the gene locus. HLAII alleles in polyallelic samples are counted individually, the same as monoallelic samples. **C** Data points (peptide-allele(s) pairs) are clustered by the peptide sequence similarity. The number and percentage of data points that fall into six arbitrary cluster size categories are shown.

## HLAIIPred architecture mimics the biological process of pHLAII presentation

Figure 2 depicts the main components of the HLAIIPred model architecture. Briefly, the HLAIIPred used a transformer-based architecture, comprising of two encoder blocks that encode the input peptide sequence and HLAII alleles. The outputs of the two encoder blocks are then fed to another encoder block, interaction encoder, to model peptide-HLAII interaction. This cross-attention block is followed by a feed-forward neural network that predicts a presentation likelihood for each peptide 9mer in peptide-allele pairs. The final presentation score is determined as the maximum likelihood among all pairs. Below is a description of the main components of the model:

**Peptide Encoder:** The input peptide sequence is first encoded into a sequence of continuous representations by a trainable embedding layer. The embeddings are then passed through a stack of Transformer's encoder blocks to capture the long-range dependencies between the AAs. We next extract all possible 9mer sequences from the encoded peptide sequence with a sliding window of 9 AAs, thus enumerating all potential 9mer cores, while their AA representations are learned from the entire peptide sequence.

**Allele Encoder:** In this block, we first represent each HLAII allele by the pseudo sequence of its alpha and beta chains, based on the work of the NetMHCIIpan team[9]. The concatenated pseudo sequences for each allele are then embedded into a continuous representation and subsequently passed through a stack of Transformer encoder blocks. Self-attention mechanism in the encoder blocks captures the dependencies between the embedded AAs of the alpha and beta chains of the HLAII allele in the binding groove.

**Interaction Encoder:** The main component of this block is the cross-attention mechanism, which is implemented similarly to the original encoder-decoder attention layer in the Transformer model. For all possible 9mer-allele pairs, the encoded alleles provide the memory key and value vectors, while the encoded 9mers provide the query vectors. This allows every encoded AA in a 9mer to attend to the encoded AAs in the corresponding alpha and beta chains of the HLAII allele. The Interaction Encoder module is also composed of a stack of identical layers, comprising attention layer, with residual connection, followed by a normalization layer.

## HLAIIPred outperforms other existing models in identifying presented peptides in monoallelic samples

We evaluated the performance of HLAIIPred on our hold-out test set, and report the F1 score of 0.83, area under the receiver operating characteristic curve (ROC-AUC) of 0.92, and precision-recall AUC (PR-AUC) of 0.92. These results confirm that HLAIIPred is able to predict pHLAIIs with high accuracy. Since our test set may have overlap with the training set of the other existing models, we prepared an independent dataset to allow a fair comparison. We used the recently published monoallelic HLAII Immunopeptidomics dataset from Stražar et al.[17]. Chronologically, this dataset should not be included in the training of any of the existing models that are published recently. Any peptide that shares a 9mer with peptides in the HLAIIPred training dataset is also removed and recycled to the training data if it was non-redundant. Consequently, we curated a dataset of unique peptide ligands for 42 unique alleles, including 20 HLA-DQ, 13 HLA-DP, and 9 HLA-DR. These alleles appear less frequently in the available datasets, and thus can provide a solid benchmark for the out-of-domain applicability of models. Additionally, we generated new random decoys for this dataset to report the classification performance of the models (as described in the Methods section).

We compared the performance of HLAIIPred with Graph-pMHC[25], MixMHC2pred-2.0[16], and NetMHCIIpan-4.1[11,26] on this monoallelic benchmark dataset. All these models have been developed within the past few years and are trained with a relatively up to date MHC/HLA MS-based datasets. As shown in Fig. 3A(top), HLAIIPred predicts presented peptides with greater accuracy compared to the previous models across 31 monoallelic samples. As shown in Fig. 3A (bottom), 10 alleles (highlighted in red) are highly underrepresented in the training dataset, with less than 1% of the total size. Among those 10 alleles, HLAIIPred outperforms all other models in predicting pHLAIIs for 8 of them, while Graph-pMHC is better than HLAIIPred for only 2 alleles. Notably, the true positive rate of HLAIIPred for all 42 alleles is 69%, while it is 58% for Graph-pMHC, 54% for MixMHC2pred-2.0, and 35% for NetMHCIIpan-4.1. This demonstrates an increase of at least 16% in prediction of presented peptides. We further tested the ability of different models in distinguishing presented peptides from random decoys using other classification metrics. As shown in Fig. 3B, HLAIIPred significantly outperforms other models, achieving an overall F1 score of 0.77, compared to 0.71 for Graph-pMHC, 0.69 for MixMHC2Pred-2.0, and 0.51 for NetMHCIIpan-4.1. HLAIIPred shows a slight improvement over other models in terms of ROC-AUC and PR-AUC as well.

In order to investigate the HLAIIPred design choices, we examined the agreement between HLAIIPred cross-attention and contact maps for 50 pHLAII complexes (see Fig. 3C, D). Figure 3D illustrates that the percentage of contact maps recovered by top-3 HLAIIPred attention values ranges from 20% to 50% for alpha chains and 10% to 40% for beta chains. Given that the models are trained solely on peptide and allele sequences, these numbers show that the attention heads are able to recover a significant portion of the structural contact maps. In addition, we analyzed the effect of cluster weights on the model performance by training two versions of the model: with and without incorporating cluster weights during training. Figure 3E shows that the ROC-AUC of test dataset is evenly distributed across different cluster weights when cluster weights are applied in the loss function. Conversely, the ROC-AUC favors larger clusters (i.e., subset of data with smaller cluster weights) when cluster weights are not considered during training.

## Immunogenicity assessment of therapeutic antibodies with HLAIIPred

We evaluated the utility of HLAIIPred in immunogenicity assessment of therapeutic antibodies following an approach similar to previously reported by several groups[25,32,33]. This approach seeks to find a correlation between the number of immunogenic CD4 T cell epitopes in the antibody sequence and its overall immunogenicity. Immunogenic epitopes are regions of a protein sequence that can be recognized by T cells, induce their activation, which in return can activate drug-specific B cells and lead to their differentiation in anti-drug antibody (ADA)-secreting plasma cells[34]. Here, we simplified the definition of epitope as a peptide sequence that is predicted by HLAII presentation models. To compensate for such simplification, peptides that are present within any human antibody germlines, which could be ignored by the immune system or activate regulatory T cells, are filtered out[35] (see Methods for details). This allowed to focus on predicting non-human epitopes that are more likely to trigger an immune response. In order to extract all possible peptides from the antibody sequences, we used a sliding window with length 9 and 12-19 AAs. The choice of the window size is based on the prediction performance of the HLAIIPred for peptides with different lengths (see Supplementary Information Figure 1 and 2). Ultimately, only the non-germline 9mer cores of presented peptides are considered as epitopes. To consider the allelic distribution in the human population, we used nine representative common DR alleles as reported by clustering of HLAII alleles in Southwood et al.[36] and Lund et al.[37]. For each antibody sequence, we counted the number of unique non-germline pHLAII cores predicted by HLAIIPred, and used this number as an input feature to predict the immunogenicity.

We used a benchmark dataset of 217 therapeutic antibodies along with their antidrug-antibody (ADA) responses as reported by Marks et al.[38]. We further refined the data and identified 210 unique antibodies with their corresponding immunogenicity labels (160 non-immunogenic, and 50 immunogenic), as described in the Methods section. We ran MixMHC2pred-2.0, Graph-pMHC and NetMHCIIpan-4.1 along HLAIIPred on this dataset to compare their respective performance. Fig. 4A shows the epitope count distribution for the immunogenic and non-immunogenic antibodies for each model. The classification performance is

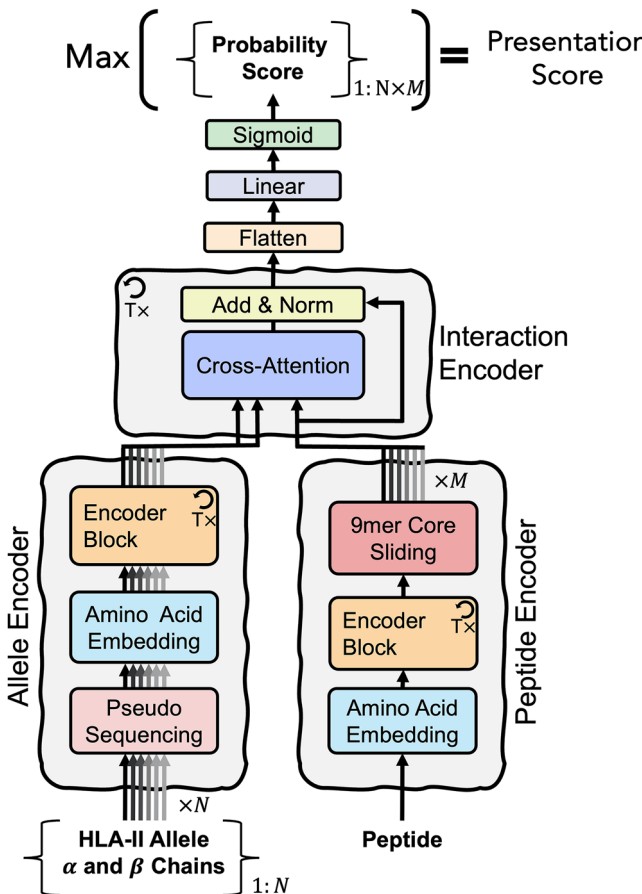

**Fig. 2 | The architecture of the HLAIIPred model.** The input data includes a peptide sequence and one or more HLAII alleles ($N$ in total). The model extracts $M$ 9mers from the encoded peptide sequence with a sliding window of 9 amino acids. Probability score for each 9mer-allele pair ($N \times M$ in total) will be predicted as the immediate output of the model. The final presentation score is determined as the maximum score among all pairs.

reported in terms of the ROC-AUC and the precision-recall curve (PR-AUC). The results in Fig. 4B show that ROC-AUC achieved by HLAIIPred is very similar to other models. However, the PR-AUC is a better suited metric for such imbalanced dataset, and showed that HLAIIPred outperforms other models by at least 5% (0.598 compared to the 0.565).

Next, we sought to find the best cutoff value for the epitope count to distinguish between immunogenic and non-immunogenic antibodies. Using a leave-one-out cross-validation strategy (see Methods for details), we found the following cutoff values for the best performance of each model: 35 for HLAIIPred, 23 for MixMHC2pred-2.0, 36 for Graph-pMHC, and 13 for NetMHCIIpan-4.1 (also depicted in the Fig. 4A). The F1 score using the best cutoff value is reported for each model in Fig. 4B. HLAIIPred demonstrated a 2% relative increase with respect to the second-best model. In the context of therapeutic proteins and the potential severe consequences on safety and efficacy of ADA, would they develop, a small improvement in immunogenicity prediction can have a significant impact on the development of therapeutic antibodies, by reducing the risk of loosing benefit to patients.

## HLAIIPred can be reliably applied to antibody engineering

Although clinical immunogenicity prediction is extremely valuable to rank multiple candidate leads in a program, it does not provide the necessary information to mitigate the risk through re-engineering or designing low immunogenicity risk candidates in the first place. Therefore, we investigated whether HLAIIPred could be applied to identify such immunogenic hotspots and help to de-risk by engineering out those potential liabilities. We

performed this evaluation using experimental pHLAII presentation data, generated using an MHC-associated protein proteomics assay (MAPPs)[39,40] for bococizumab. Bococizumab is a notable Pfizer-developed monoclonal antibody that has been shown to be highly immunogenic in clinical trials.

We designed a similar pipeline as described in the previous section and illustrated diagrammatically in Fig. 5A to predict pHLAIIs in each antibody (see Methods for details of our strategy). Figure 5B shows the heatmap of the predicted non-germline hotspots along bococizumab sequence for each model, and a comparison of the hotspots predicted by HLAIIPred and experimentally identified pHLAIIs. As expected, the non-germline pHLAIIs predicted by all models and identified by MAPPs are located in the variable domains. The only exception is one segment of the heavy chain that is located by Graph-pMHC but is not present in the experimental data.

## HLAIIPred can prioritize vaccine neoantigens and predict binding specificities of HLAII alleles

HLAIIPred could also be used to predict the presentation of neoantigens by HLAII molecules and prioritize the peptides that are likely to elicit a T cell response. We use a set of 125 personalized melanoma vaccine neoantigens with corresponding immune response data from the work of Sahin et al.[2]. Most of these peptides have a length greater than 26 AAs, which is the maximum length of the peptides in our training data. Thus, we used the sliding window approach to extract shorter sequences, in which the most likely binding cores will be selected by the model. Each peptide will be assigned a presentation score based on the maximum score among all selected 9mer cores and the nine common DR alleles. Results show that 85.3% of the selected neoantigens with post-vaccination immune response are correctly predicted with a score above HLAIIPred's cutoff (0.5). We then categorized the data based on their predicted scores to three groups: above 0.95 (high-likelihood), between 0.5 and 0.95 (low-likelihood), and below 0.5 (not-presented). Figure 6A illustrates the distribution of the immune response for each group. Among peptides with high-likelihood scores, 77.8% of them elicit a T cell response after vaccination, while this number is 61.0% for peptides with low-likelihood scores. These results not only show that HLAIIPred can be used to prioritize neoantigens for the vaccine design, but also confirm that higher presentation scores are associated with a higher likelihood of immune response.

We then investigated the ability of HLAIIPred to predict binding specificities of HLAII alleles. We performed this evaluation using 51 crystal structures of the pHLAII protein complexes from the Protein Data Bank (PDB). All these structures are also included in the discovery and test sets of a previous study by Mikhaylov et al.[19]. The authors reported that the core residues of peptides that interact with HLAII molecules can be predicted with high accuracy using their proposed AlphaFold-based[41] pipeline, called TFold (miss-predicting only 2 out of 51 structures). We used their polished data and binding core annotation as the ground truth to evaluate the performance of HLAIIPred and other sequence-based pHLAII presentation models for the binding core identification. The results show a significant drop in the performance of the sequence-based models in comparison with the TFold structure prediction (see Fig. 6B). However, HLAIIPred achieved better performance compared to other existing methods, with missing only 10 out of 51 pHLAII structures. HLAIIPred exhibited the lowest number of errors among DQ and DP alleles, however, missed more binding cores in contact with DR alleles compared to the Graph-pMHC. We argue that the application of sequence-based pHLAII models remains justifiable, taking into account the computational cost of the TFold pipeline, which takes minutes to run compared to milliseconds for HLAIIPred on the same GPU device.

Considering the acceptable performance of HLAIIPred in predicting the binding core residues, we used the predicted 9mer cores to identify the binding specificities of HLAII alleles. We first extracted the core residues of peptides from the monoallelic dataset from Stražar et al.[17]. For each allele, the Shannon entropy of the 20 AAs in each of 9 positions of the binding cores of the presented peptides is calculated. A lower entropy value indicates a higher binding specificity of the allele for the corresponding position. As

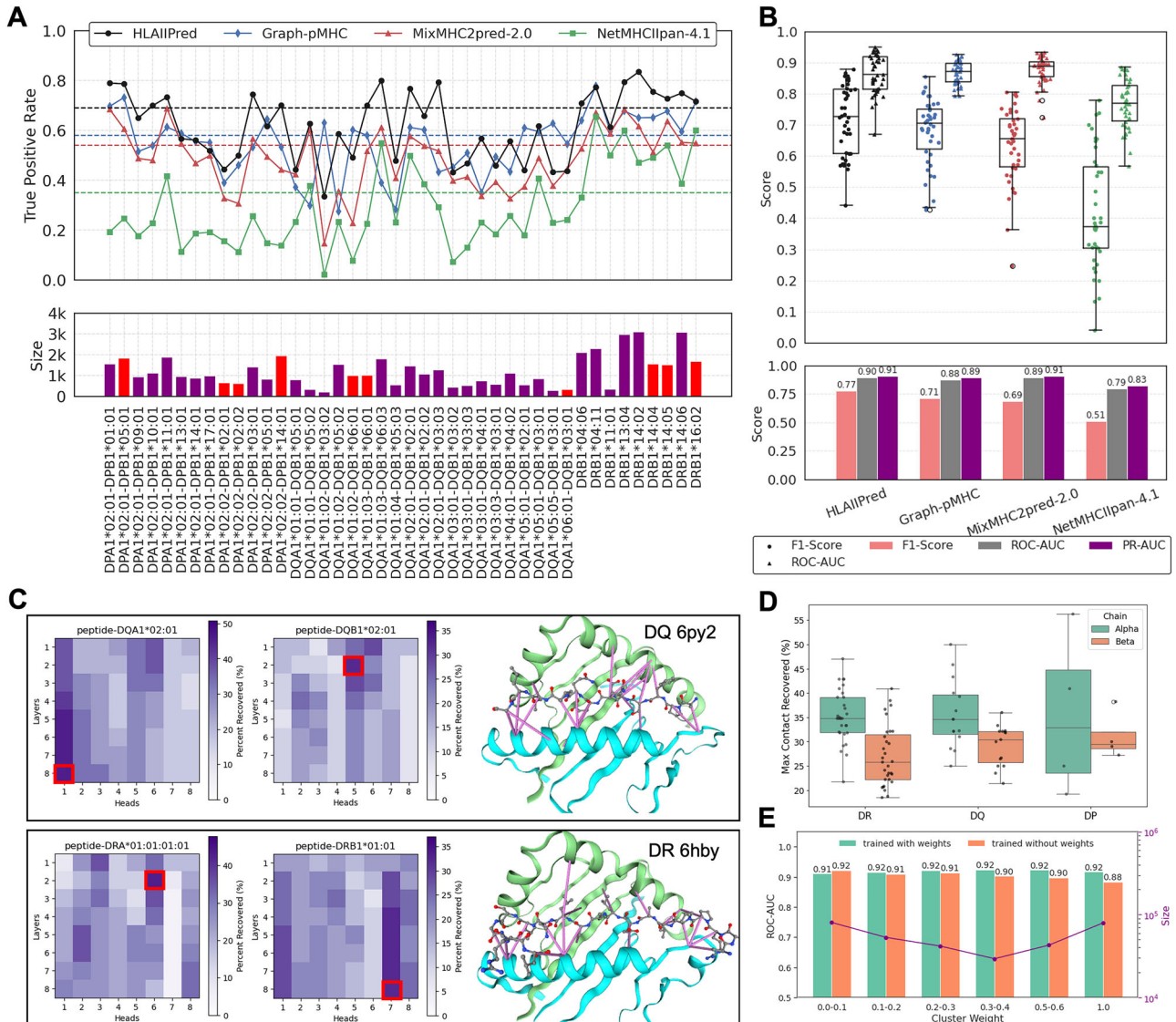

**Fig. 3 | Performance of HLAIIPred and other existing models on an independent monoallelic dataset from Stražar et al.[17]. A** top: The ratio of presented peptides in each monoallelic sample that are correctly predicted by each model. Dashed line shows the overall true positive rate for each model. bottom: the size of each allele sample. The red color indicates alleles that are overly underrepresented (less than 1%) in the training dataset. **B** top: The distribution of F1 scores and ROC-AUC per monoallelic sample for distinguishing presented peptides from random decoys, grouped by model. bottom: The bar plot of overall F1-Score, ROC-AUC, and PR-AUC for each model. **C** Agreement between attention and contact maps for two

pHLAII complexes (PDB ID: 6py2 and 6hby) across different layers and heads of interaction encoder. The recovered contact maps are shown for the best layers/heads (indicated by red squares) using purple lines between peptide (ball and stick representation) and HLAII protein (cartoon representation[53]). **D** The maximum percentage of contact maps recovered by HLAIIPred attention mechanism for 50 pHLAII complexes. The contact maps are defined using crystal structures and the distance cutoff of 5 Å. **E** The ROC-AUC of subsets of test dataset grouped by different cluster weights for two HLAIIPred models, once trained with cluster weights and once without cluster weights.

depicted in Fig. 6C, DR alleles showed lower entropy values at positions 1, 4, 6, and 9, which are the anchor positions for peptide binding to the HLAII molecule. DP alleles exhibited AA-binding preferences at positions 1 and 9, but also at positions 4,6, and 7 to a lesser extent. The anchor positions 7 and 9 for the HLA-DP are the most preferable positions among the three gene loci. DQ alleles are more indiscriminate in their binding preferences, but showed lower entropy values uniquely at position 3 and to a smaller degree at positions 4 and 6.

## Discussion

The prediction of peptide presentation by HLAII molecules is a technically challenging problem, requiring the best of model development and data preprocessing strategies. We leveraged the Transformer's encoder to capture the long-range dependencies between AAs in each sequence. The cross-attention mechanism also enables the pairwise interaction between peptides

and HLAII alleles. The HLAIIPred architecture allows us to learn the presentation of peptides by specific alleles based on the highest score among all pairs of binding cores and alleles. Unlike the recent graph neural network models, the core-allele pairs are not enumerated from the beginning, forming isolated subsamples of the pHLAII complex. Instead, our model encodes each of input sequences iteratively, and learn the importance of each AA and its relative position with respect to all other AAs in that sequence. As a result, we observe that our model does not benefit from any additional data from flanking residues at the cleavage regions, thus reducing the chance of incorrect cleavage assignment, which has its roots in the lack of one-to-one mapping between peptides and proteins, or the lack of cleavage site information in the data (for example for neoantigen prediction). Another advantage of our model is the use of attention mechanism, where AA representations are not only informed by their adjacent residues, but they are connected to all other residues in the sequence as well. In contrast in

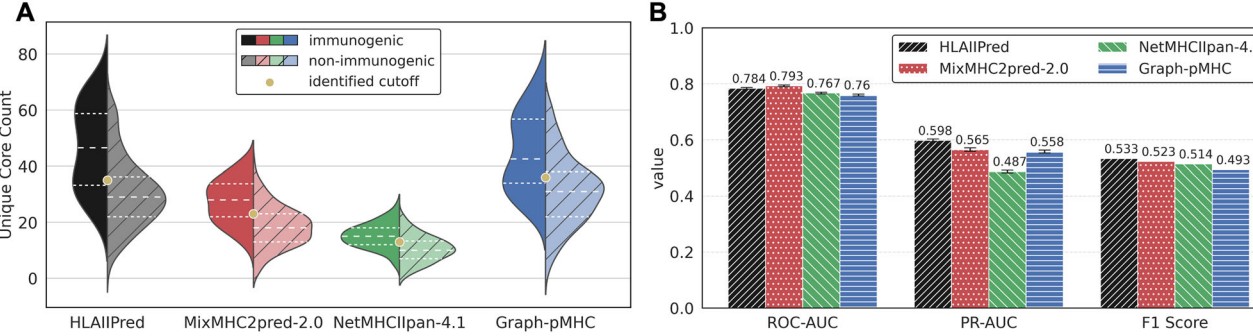

**Fig. 4 | Application of pHLAII presentation models for the immunogenicity prediction. A** Distribution of 210 immunogenic and non-immunogenic antibodies based on the number of unique cores (epitopes) predicted by each model. **B** The performance of each model in terms of the ROC-AUC and PR-AUC for the

immunogenicity prediction (The error bars are based on leave-one-out analysis). The F1 score is reported for each model based on the best cutoff value for the epitope count on the leave-one-out setting.

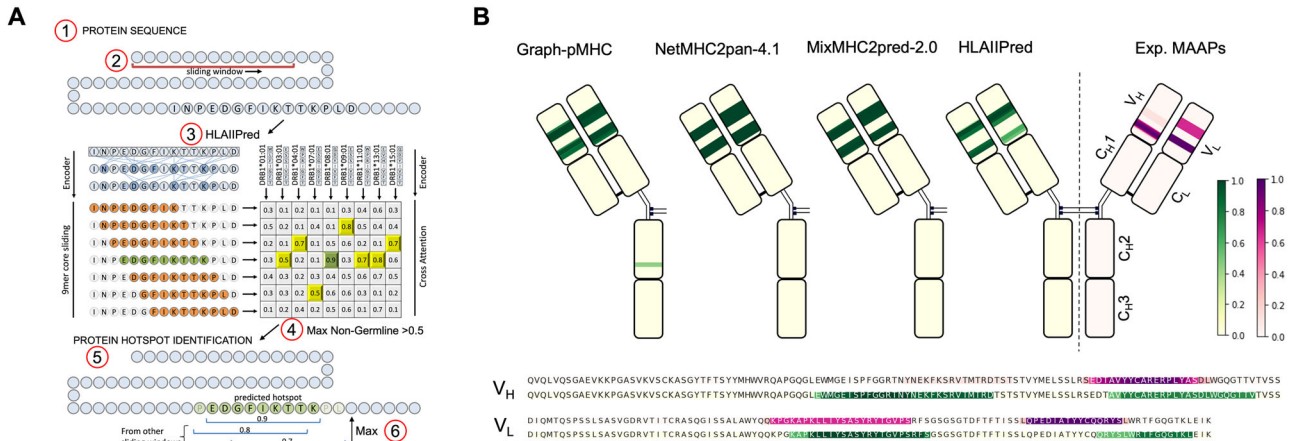

**Fig. 5 | HLAIIPred identifies pHLAIIs in the Pfizer antibody library in line with the MS-identified peptides. A** The schematic of the antibody sequence screening pipeline. **B** The heatmap of the predicted AA presentation scores for the

bococizumab antibody. At the bottom, the heatmaps of the experimental pHLAIIs and HLAIIPred predictions in the heavy and light chain variable domains are illustrated.

graph-based models, the intramolecular message passing is only allowed with the adjacent residues, and intermolecular connections are also restricted by predefined graphs. The fully-connected models also lack any direct connection between the residues, and they only rely on an abstract combination of the learned features from all residues. Thus, a customized model architecture that is designed based on the biological process of peptide presentation, and leverages the best of the deep learning models, can significantly improve the prediction performance.

In terms of data preprocessing, we noticed that clustering of peptides based on sequence similarity can prevent the model from overfitting to dominant binding cores. Given the nature of data, pHLAII models must infer intermediate knowledge, like the core residues of peptides and the most likely HLAII alleles, while such information is not explicitly provided in data. To manage this, it is essential to implement regularization techniques that prevent the model from memorizing overrepresented patterns. Additionally, the utility of monoallelic data is crucial to guide the model to learn the presentation and binding specificities of peptides by specific alleles. By employing these data-centric strategies, our model not only demonstrated robustness across diverse HLAII alleles, but also outperformed the other existing models on a monoallelic dataset with less frequent alleles, underscoring its generalizability.

The application of HLAIIPred for the antibody immunogenicity prediction is a promising step towards the development of non-immunogenic therapeutic antibodies. We provide comparison data for bococizumab, a monoclonal antibody, with known high clinical immunogenicity.

Experimental data showed that the non-germline pHLAIIs are located in the variable domains of this antibody, and pHLAII presentation models can recover these hotspots with high fidelity. Thus, our model can be potentially employed in the early stages of the antibody discovery to redesign such antibodies and reduce the risk of immunogenicity. For future work in this direction, we envision the integration of the antibody engineering and the immunogenicity prediction in a closed loop. This can be achieved by using the predicted hotspots to guide the design of non-immunogenic antibodies, and then experimentally validate the predictions to improve the model performance and derisk therapeutic antibodies.

One of the limitations of the current study is the lack of robust training strategy for binding core identification. This is mainly due to the lack of 3D structure for the pHLAII complex in the training data. Two recent studies have shown promising performance for the structure prediction of pHLAII complexes[19,42]. Both are AlphaFold-based pipelines, and demonstrated that the core residues of peptides that interact with HLAII molecules can be predicted with high accuracy. However, these models are not able to distinguish between the presented and not-presented peptides, unless they are directly fine-tuned on the immunopeptidomics data[42]. Considering the classification accuracy of such models and their computational cost, we believe the sequence-based approaches are still more practical for high-throughput screening of proteins and antibodies. However, the structure-based models can be used as a complementary tool to the sequence-based models to improve the generalizability[43] and prediction of the binding core residues. Our model shows that we can predict the binding core registers

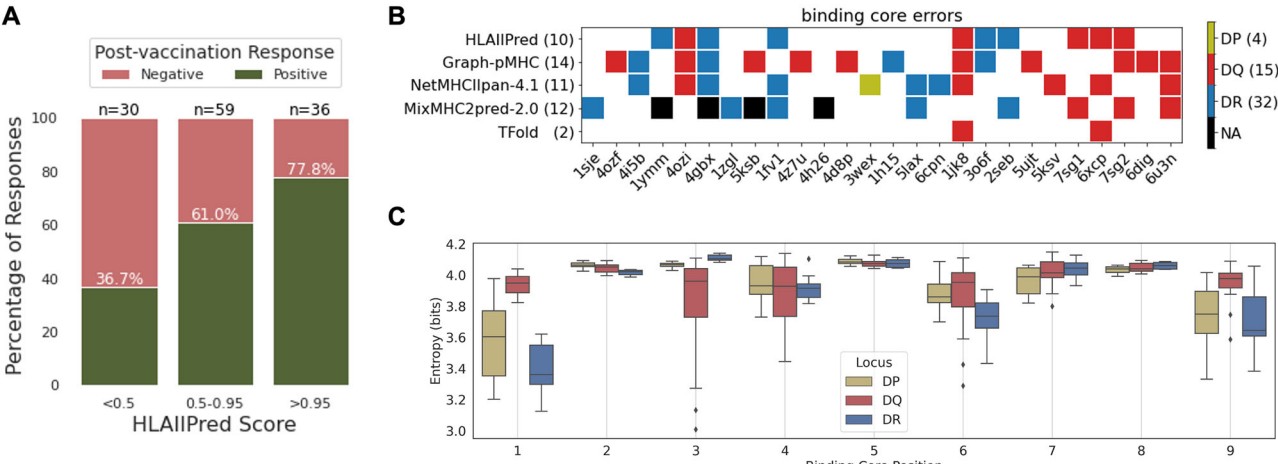

**Fig. 6 | HLAIIPred can prioritize peptides with the post-vaccination T cell response, and predicts the binding specificities of HLAII alleles. A** The distribution of the immune response for 125 personalized melanoma vaccine neoantigens based on the presentation scores predicted by HLAIIPred. The peptides are categorized based on their scores to three groups: above 0.9 (high-likelihood), between 0.5 and 0.9 (low-likelihood), and below 0.5 (not-presented). **B** The diagram

of the binding core errors (filled squares) made by each model (rows) for the pHLAII structures with the corresponding PDB IDs (columns). We collected 51 structures in total, and this plot only shows the PDB IDs for the structures that have at least one error by any of the models. NA is assigned when a peptide length is not supported by the model. **C** The Shannon Entropy of the 9mer binding cores at each position, presented by different HLAII alleles and grouped by their gene locus.

with higher accuracy compared to the other sequence-based models (Fig. 6B). Nevertheless, the pMHCII structure prediction pipelines such as TFold[19], have reported higher accuracy in the prediction of the core residues for samples with relevant templates. We are currently working on the integration of the structure-based models with HLAIIPred to guide the prediction of the binding cores along with the presentation likelihood of peptides.

Prediction of pHLAII presentation is an important first step towards the prediction of the immune response. However, the next step to increase the confidence in such predictions, is to predict the recognition of the presented peptides by T cell receptors (TCRs). HLAIIPred has shown potential applications to peptide-based cancer vaccine design, particularly to the recognition of neoantigen by CD4+ T cells. The model, however, can benefit from the training data that includes recognized pHLAII complexes by TCRs. We are now seeing the emergence of such models using both sequence-based and structure-based approaches[44,45]. The integration of these datasets with our model development efforts will provide a comprehensive tool to identify peptides with the highest potential to induce a T cell response. Addressing these areas by building such tools will ultimately contribute to the development of safer and more effective vaccines and antibody drugs for cancer, infectious diseases, and autoimmune disorders.

## Methods
### Data Preparation
We compiled the MS-based experimental data from Racle et al.[16] and HLA Ligand Atlas[46]. Each of these datasets aggregates MHC class II peptidomics data from different sources, and were combined to constitute a more comprehensive dataset. Duplicates and peptides that are not in the range of 8–26 AAs are removed. In addition, we only kept human samples that include annotated HLAII alleles. In total, we collected 597508 unique peptides from 341 samples with 172 unique alleles. However, only 54 HLAII alleles are in monoallelic samples (31%), and the rest are polyallelic samples. In total, we have 1856146 unique peptide-sample pairs, where each peptide is presented by one or more alleles. We split the data into train, validation, and test sets using 8:1:1 ratio based on the minimal 9mer overlaps and balanced gene ontology terms as described by Thrift et al.[25]. In contrast to the previous studies, we do not work with the k-fold cross-validation, and we only split the data once. The training data is used to train the model, the validation data is used for hyperparameter tuning and the selection of best model, and the test data is used to evaluate the performance of the best

model. For the clustering, we use easy-cluster workflow from MMseqs2[47,48] with the minimum sequence identity of 0.8. We tested different threshold and selected 0.8 based on the performance of the model on the validation set. Each row of the training data will be assigned a weight based on the inverse of the cluster size in the full training set. These weights are used in the loss function as described in the next section to prevent the model from memorizing the dominant binding cores.

We randomly selected peptides from the human proteome to add negative samples (aka, random decoys) to each split. In order to increase the coverage of the human proteome, we generated a new set of random decoys for each training epoch. We considered two scenarios for the random decoy generation which resulted in two different trained models: 1) random decoys are randomly selected from the human proteome, and 2) random decoys are randomly selected from the human proteome and screened to avoid any 9mer overlap with the presented peptides. Regardless of which scenario is used, the length distribution and number of the negative data remains the same as the positive data at each epoch. At the inference, we average the predictions of the two models to get the final presentation score.

### HLAIIPred Model Implementation
The HLAIIPred model is implemented in PyTorch[49] deep learning framework. The architecture, as schematically depicted in Fig. 2, is comprised of 8 Transformer encoder blocks (T=8) in the peptide and allele encoders. Each encoder block consists of a multi-head self-attention mechanism with 8 heads. The size of latent features is fixed to 128 in all layers, including the embedding layer. The interaction encoder is also composed of 8 cross-attention blocks with 8 heads, followed by a stack of linear layers and layer normalization. The output of the final interaction block is flattened to produce a 1152-dimensional feature vector, which is then passed through a dropout layer[50] with a dropout rate of 0.1, and finally to the output layer comprising a single node with Sigmoid activation function. The output of the Sigmoid function is in range [0, 1], and it represents the likelihood of the 9mer cores being presented by each HLAII allele in the input. Thus, the cutoff value for the presentation score is 0.5, because we always keep the number of positive and negative samples balanced in the training data. The final presentation score is calculated as the maximum score among all pairs of 9mer cores and alleles. The only non-linear functions in this model are rectified linear activation unit (ReLU) and the final sigmoid activation function. We set the dropout rate to 0.1 in the embedding, attention, and residual linear layers. A custom dropout layer is implemented to randomly

drop predicted scores for half of the alleles in the polyallelic samples, improving generalization to underrepresented alleles.

Peptide inputs are padded to the maximum length of 30 AAs. The model is trained with a maximum number of 14 alleles per sample. The network was trained by minimizing Binary Cross Entropy loss using AdamW optimizer[51] with default settings. The loss function uses the pre-calculated weights for each sample based on the cluster size of the presented peptides. Random decoys will also get the same weight as the presented peptides that they are linked to. The learning rate is set to 1e-5, and the model is trained for 50 epochs. The learning rate is decayed by a factor of 0.7 if the validation loss plateaus. We stop the training if the validation loss does not improve for 5 epochs and the learning rate has decayed for 2 times. A gradient clipping of 1.0 is used to prevent exploding gradients. The models are trained with a batch size 200 on four NVIDIA V100 GPU with 32GB memory. The training of the model takes approximately 48 h.

### HLAII alleles and their pseudo-sequences

We obtain the HLAII-DR, HLAII-DP, and HLAII-DQ sequences from the PD-IMGT/HLA database[52] for all available heterodimers. We follow the same procedure as described by NetMHCIIpan-3.0[9] to generate the pseudo-sequences of the constituent alpha and beta chains. Alpha chains are converted to 15-AA long pseudo sequences, reflecting 15 residues that are most likely to interact with the peptide in the binding groove. Similarly, beta chains are converted to 19-AA long pseudo sequences. The pseudo sequences are concatenated to form the final pseudo sequence for each allele. Whenever we need to use 9 common DR alleles according to the clustering of HLAII alleles in Southwood et al.[36] and Lund et al.[37], we use the pseudo sequences of the following alleles: DRB1*01:01, DRB1*03:01, DRB1*04:01, DRB1*07:01, DRB1*08:01, DRB1*09:01, DRB1*11:01, DRB1*13:01, DRB1*15:01. Note that all these alleles share the same alpha chain, encoded by DRA1, as this gene is not polymorphic.

### Reference data for ADA immunogenicity

The full dataset for the antibody immunogenicity analysis is 217 therapeutic antibodies with their antidrug-antibody (ADA) responses curated by Marks et al.[38]. If the ADA response is above 10%, the antibody is considered as immunogenic, otherwise it is labeled as non-immunogenic. We identified 5 pairs of antibodies (i.e., 10 data points in total) with identical sequences but different ADA responses. These data differences are most likely due to different clinical conditions, such as patient-specific immune variability, if not simply due to inherent experimental or data collection errors. In 3 pairs, the ADA response of both antibodies are on the same side of the 10% threshold. Thus, we removed one of the antibodies in each pair and keep the other one with the corresponding label. In two pairs, the ADA response of one antibody is above 10% and the other is below 10%. In those cases, we removed both antibodies due to the conflicting labels. As a result, we have 210 unique antibodies with their corresponding immunogenicity labels.

In order to identify the best cutoff value, we used the leave-one-out cross-validation strategy, and divided the data into 210 discovery and validation sets. In each discovery set, the best cutoff value is selected using the ROC curve, where it simultaneously maximizes the true positive rate and minimizes the false positive rate. The best cutoff value is then used to evaluate the performance of the model on the corresponding validation set. The overall cutoff value is the average of the best cutoff values across all discovery sets. The measured cutoffs for the best performance of each model are reported in the Fig. 4A.

### Germline sequences

Our human antibody germline database was compiled from sequences downloaded from the IMGT database[52]. The germline database covers all V genes (IgHV, IgKV, & IgLV), all J genes (IgH, IgK, & IgL), and several C genes (CK, CL, CH-IgG, & CH-IgA), as well as the permuted junctions between covered J & C genes. D genes and VJ/VDJ junctions are excluded from our germline database. All unique AA sequences present in the human

IMGT database for the covered regions are included, except for those genes annotated as open reading frame (ORF) or pseudogenes.

### MAPPs Data

Experimental evaluation of HLAII peptide presentation was conducted using an internal MHCII-associated peptide proteomics assay (MAPPs) as previously described in ref. 40. Donor sets in each experiment were selected to be representative of the HLAII diversity and allele frequency of the North American population. The data for bococizumab is shown here as a representative example of the study (Supplementary Data 1A–C). The antibody used in the MAPPs assay to precipitate HLAII-peptide complexes is the anti-HLA-DP/DQ/DR antibody from ATCC IVA12 HB-145 hybridoma prepared by Kempbio Inc. Leukopaks collected from healthy donors were sourced from StemCell Technologies (Vancouver, BC, Canada) and Charles River Laboratories (Wilmington, MA, USA) in compliance with all applicable laws, regulations, and Pfizer policies. All ethical regulations relevant to human research participants were followed.

### Analysis of antibodies immunogenicity heatmaps

For each antibody sequence, the sliding window of 9 and 12–19 AAs are used to extract all possible peptides. The nine common DR alleles plus those available in donor samples are used to predict the presentation scores for each peptide. We used the maximum score among all alleles as the final score for each peptide. If the score of a peptide was above the model's cutoff (0.5 for HLAIIPred), and the predicted 9mer core did not appear in the germline sequences, the 9mer core was considered as an epitope. We assigned the epitopes score to each AA position in the antibody sequence by taking the maximum score of all 9mer cores that contain the corresponding AA. The final AA score is used to predict the immunogenic hotspots along the antibody sequence. All non-germline MS-identified pHLAIIs are aligned against the antibody sequence. The MAPPs heatmap is illustrated by the number of experimental pHLAIIs covering each AA position normalized by maximum number among all positions.

### Hyperparameters of external models

In this study, we used the MixMHC2pred-2.0, Graph-pMHC, and NetMHCIIpan-4.1 as baseline models. The recommended binary classification cutoff for each model is different. Using MixMHC2pred-2.0 and NetMHCIIpan-4.1, any peptide rated below 5% percentile are considered as presented. The cutoff value for the Graph-pMHC is zero, meaning any predicted score above 0.0 is considered as presented. If we need to use sliding windows to extract all possible peptides, the window sizes of 12-19 AAs are used for these three models. The reason for excluding 9mer peptides (compared to the HLAIIPred) is that the MixMHC2pred-2.0 and NetMHCIIpan-4.1 models are not able to predict the presentation of 9mer peptides, and the Graph-pMHC model does not recommend the use of 9mer peptides for a similar screening task. The range of acceptable peptide length for each model is as follows: MixMHC2pred-2.0 (12-21 AAs), Graph-pMHC (9-26 AAs), and NetMHCIIpan-4.1 (11-26 AAs). We provide the flanking residues to all three models as requested for their optimum performance.

### Statistics and reproducibility

The classification metrics that we use in this study are the F1 score, the area under the receiver operating characteristic curve (ROC-AUC), and the area under the precision-recall curve (PR-AUC). The F1 score is the harmonic mean of the precision and recall, and it is calculated as $2 \times \frac{precision \times recall}{precision + recall}$. We prefer F1 score to ROC-AUC and PR-AUC, because it balances the precision and recall, and requires the cutoff value for the binary classification, not leaving this decision to the users.

### Reporting summary

Further information on research design is available in the Nature Portfolio Reporting Summary linked to this article.

## Data availability

The benchmark data and HLAIIPred predictions are available at (https://github.com/pfizer-opensource/HLAIIPred). The training, validation, and test data sets are availabe at (https://zenodo.org/records/15299217). The new data generated during the current study, i.e., the MAPPs data for bococizumab, is available in the Supplementary Data 1 A-C.

## Code availability

The GitHub repository for HLAIIPred is publicly available at https://github.com/pfizer-opensource/HLAIIPred and includes Python scripts, meta data, and model weights.

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

## Acknowledgements

We would like to thank Christopher Lepsy and Laura Lin for their strategic insights, valuable feedback and discussions. This work was funded by Pfizer.

## Author contributions

M.H., N.M., T.P.H., E.B., B.R., and S.T. conceived the scientific direction, designed the experiments and analyzed the results. R.S., A.C., R.H., J.C., B.K., S.D., and S.T. provided the MAPPs data and validated the antibody screening results. M.H. and S.H. compiled the data. M.H. implemented the code for HLAIIPred and prepared the first draft of the manuscript with significant conceptual input from B.R. All authors reviewed the manuscript.

## Competing interests

The authors declare the following competing interest: all authors are or have been employees of Pfizer Inc. and may hold stock or stock options in the company.
