## [Transparent Peer Review file · Communications Biology]

HLAIIIPred: Cross-Attention Mechanism for Modeling the Interaction of HLA Class II Molecules with Peptides

Corresponding Author: Dr Mojtaba Haghighatlari

Version 0:

Reviewer comments:

Reviewer #1

(Remarks to the Author)

Review Comments for Manuscript: "HLAIIIPred: Cross-Attention Mechanism for Modeling the Interaction of HLA Class II Molecules with Peptides"

General Comments:

The manuscript presents HLAIIIPred, a deep learning model designed to predict peptides presented by HLA Class II molecules. The model demonstrates performance compared to existing methods. However, several areas require further clarification to provide a more robust context for the results.

Major Comments:

1. Challenge of Polyallelic Nature:

The manuscript discusses the challenge of polyallelic interactions, but this section could benefit from further elaboration. It is essential to highlight that the diversity of HLA Class II alleles can lead to polyallelic presentations. Specifically, only exon 2 contributes to the peptide-binding domain, meaning it may not be necessary for the entire HLA Class II gene to be monoallelic. Additionally, DRB1/3/5 are paralogs, implying that monoallelic expression is not inherently required. The methods section should be more explicit about including peptides presented by multiple alleles in the analysis.

2. Statistical Comparisons:

The manuscript would benefit from a more comprehensive statistical comparison with other existing tools. Including additional performance metrics at ALL times—such as recall, precision, and ROC-AUC—would provide a clearer context for how HLAIIIPred compares to existing methods.

3. Data Availability and Reproducibility:

To ensure transparency and reproducibility, it is important to make the curated dataset and full analysis code publicly available. This would allow other researchers to validate, ensuring the model's results are robust.

4. Peptide Length Considerations:

Most importantly, while the model's ability to include peptides with varying lengths, there are concerns about potential bias towards shorter peptides that might artificially inflate performance. The manuscript should provide more detailed data on the distribution of peptides analyzed across different length categories. In particular, the selection of 9-mer peptides for MHC Class II binding should be well-justified, as most literature suggests that MHC Class II typically accommodates longer peptides (i.e., peptides longer than 10 amino acids). A stratified performance comparison based on peptide length would help address this concern.

Minor Comments:

1. Typos:

There are typos throughout the manuscript, such as "DQA101:01DQB105-02" and "50This". Please carefully check.

2. Inconsistent Information:

In the results section, the manuscript states that the true positive rate (TPR) of HLAIIIPred for all 42 alleles in the test set is greater than 50%. However, this is inconsistent with the data presented in Figure 3A, where not all alleles with a TPR above 0.5. Additionally, the abstract mentions a "16% increase in prediction of presented peptides compared to the second-best model on a set of unseen peptides presented by less frequent alleles," but I could not find the corresponding information in the main text. This discrepancy should be addressed to ensure consistency.

3. Definition Clarity:

In the reference section, specifically in the data for ADA Immunogenic, it would be helpful to define the term "pair" more clearly to improve readability. Additionally, it would be helpful to clarify how possible antibodies with identical sequences can exhibit different ADA responses. This clarification would help readers better understand the context of the results.

Conclusion:

While HLAIIIPred presents promising advancements in predicting peptide-HLA interactions, addressing these comments will significantly enhance your work's clarity, accuracy, and impact.

Reviewer #2

(Remarks to the Author)

The authors present a peptide-MHCII sequence-based binding predictor exploiting transformers and cross-attention. They compare their model against state-of-the-art predictors on peptide presentation and immunogenicity assessment over four experiments. The work includes a well-needed data-focused approach and attention to the out-of-distribution performance issues. The manuscript is well-written, the figures are good quality and the provided code is clean and easily understandable.

My concerns are mainly focused on reproducibility and AI explainability.

Major concerns:

1. In the monoallelic sample comparison experiment, while mentioned by the authors, the actual overlap between train and test set alleles distribution is not shown. To properly understand the generalization capability of HLAIIIPred (and compare it with the other mentioned predictors), it is important to know the actual distance of the test set from the training set. I suggest the authors add a figure (supplementary would be fine) showing the abundance of each allele from the dataset they curated from Strazar et al. compared with the same alleles abundance in their training set, and (only if possible) also the training sets used by the other software the authors compared with. I suggest the authors use the alleles pseudosequences when checking which alleles are "missing" from the training set, instead of simple allele type names that might change for mutations in non-binding-crucial parts of the MHC sequence. Pseudosequences can also be used to calculate the average or minimum distance of each "missing" allele from the training set. In this way, a reader can understand how much the training set(s) and architecture(s) contribute to the performances shown in Fig 3A.

2. I highly recommend the authors release the final data they used in this work. While it is great to have the code available on GitHub, parsing codes can have issues and might no longer work when a source database is updated. For this reason, releasing csv/tsv/xlsx files containing the training and test datasets (from public sources) used in this work is important for reproducibility and results interpretation.

3. In the discussion, the authors suggest that HLAIIIPred is capable of learning intermolecular interactions for single AAs/positions as opposed to fully-connected models which only rely on abstract combinations of features. Is it possible then for the authors to plot the attention weights of HLAIIIPred, maybe in a supplementary figure, to support this point? If HLAIIIPred is learning biologically significant interactions, this should be reflected in the attention weights. These should show strong attention between, for instance, the anchor residues of a peptide and the pocket-forming residues they interact with on the MHC side.

Minor concerns:

1. In the first section of the results, the authors mention that weights from peptide clusters are used to modify the loss function. This sounds like a great idea in theory, but I wonder how much it is actually affecting the results. From the model training history provided, it's noticeable that in the first epoch of training, HLAIIIPred already shows a remarkably low loss of 0.1, given the task.

I think this work could largely benefit from training HLAIIIPred using only the BCE loss (i.e. without multiplying it with the peptide clusters weights) and showing/reporting the difference in initial loss and final model performances. If the loss should significantly increase upon removing the clusters' weights, the authors would have a stronger point on how their data-centred approach resulted in a better predictor.

I understand training HLAIIIPred takes a long time, so I leave this as optional to the authors.

2. In the Introduction "Polyallelic nature of the data", why would the availability of more monoallelic data magnify the challenge of negative samples? Monoallelic data will provide just as many negative samples as multiallelic data (none). Do the authors mean here that having more MS data (independently of these being mono- or multi-allelic) increases the imbalance between positives and negatives, simply because there are no negatives? If so, I suggest rephrasing this sentence, as now it seems to imply that monoallelic samples specifically provide less negative data than multiallelic ones do.

Typos and miscellaneous:

1. Typo: Introduction, first paragraph, "from a genomic perspective"

2. In the methods, when describing HLA-II pseudosequences, NetMHCIIpan-3.0 is missing the reference.

Version 1:

Reviewer comments:

Reviewer #1

(Remarks to the Author)

Reviewer #2

(Remarks to the Author)

The authors extensively replied to all my comments and questions, and provided very interesting additional results. The publication of train, validation and test data allows for better reproducibility and analysis of this work. I just have one more comment, but I trust the authors can adjust the text without the need for a new round of reviews. Moreover, I encountered a small issue with the github/zenodo links, explained below.

Besides this, I have no further comments, and I wish the authors a swift path to publication with this very nice work.

Major comment:

In the last section of the results, "HLAIIIPred can prioritize vaccine neoantigens and predict binding specificities of HLAII alleles.", the authors defend the applicability of sequence-based approaches over structure-based approaches with multiple arguments, including "the poor performance of structure prediction models in terms of distinguishing between the presented and not-presented peptides". I do not think this conclusion is fair to the article the authors cite, which achieved almost exactly netmhcpan's performances (1% difference) by adding a small predictor on top of alphafold. I would not refer to such performances as "poor", especially considering they rely on a general-purpose modeling tool like Alphafold, which can be easily adapted to minimize the errors it makes when predicting pMHC structures. Moreover, structure-based methods have shown larger generalization power than sequence-based ones (<https://www.nature.com/articles/s42003-024-07292-1>). Similarly, having limited structural templates only poses an issue when trying to predict binding cores because it limits the benchmark set we can use for testing, but the available templates are more than enough to generate large libraries of good 3D models (showcased by softwares like PANDORA or APE-Gen2.0). I would rather encourage the authors to focus their point on the speed and thus large-scale applicability of sequence-based methods, which remain vital when screening large patients mutanomes. I can see the performance gap being filled very soon, quickly rendering such point moot, but it will be far more challenging for structure-based methods to achieve similar speeds than sequence-based ones. The authors could strengthen this point by adding a mention to the time taken by each tool on one pMHC-II case: ~10 minutes in tfold (on a A100 GPU) and probably less than a second for HLAIIIPred.

Minor:

The links to github and zenodo do not work properly. The github link should be "<https://github.com/pfizer-opensource/HLAIIIPred>" but the one in the manuscript now is "<https://github.com/pfizeropensource/HLAIIIPred>" without a dash, which returns 404.

Similarly, if the zenodo link is clicked on instead of manually copy-pasted, it includes the fullstop at the end ("[https://zenodo.org/records/15299217.](https://zenodo.org/records/15299217)") which directs to an empty page. Just adding a space between link and fullstop will solve this issue.

Reviewer #1 (Remarks to the Author):

Review Comments for Manuscript: “HLAIIPred: Cross-Attention Mechanism for Modeling the Interaction of HLA Class II Molecules with Peptides”

General Comments:

The manuscript presents HLAIIPred, a deep learning model designed to predict peptides presented by HLA Class II molecules. The model demonstrates performance compared to existing methods. However, several areas require further clarification to provide a more robust context for the results.

Major Comments:

1. Challenge of Polyallelic Nature:

The manuscript discusses the challenge of polyallelic interactions, but this section could benefit from further elaboration. It is essential to highlight that the diversity of HLA Class II alleles can lead to polyallelic presentations. Specifically, only exon 2 contributes to the peptide-binding domain, meaning it may not be necessary for the entire HLA Class II gene to be monoallelic. Additionally, DRB1/3/5 are paralogs, implying that monoallelic expression is not inherently required. The methods section should be more explicit about including peptides presented by multiple alleles in the analysis.

Thank you for highlighting this important point. The discussion on the challenges of polyallelic data was intended to focus solely on ML modeling rather than the nature of the immune response. We have added clarifying sentences in the Introduction,

“Each gene locus encodes thousands of alpha and beta chains that will be combined to form heterodimer proteins with diverse peptide presentation specificities, which is essential for the immune system to recognize a wide range of antigens.”,

and in the corresponding paragraph,

“Although polyallelic data increases the diversity of HLAII alleles in response to antigens, it complicates the training of machine learning models due to the non-deterministic mapping between alleles and presented peptides in training data.”,

to clarify this point further.

2. Statistical Comparisons:

The manuscript would benefit from a more comprehensive statistical comparison with other existing tools. Including additional performance metrics at ALL times—such as recall, precision, and ROC-AUC—would provide a clearer context for how HLAIIPred compares to existing methods.

We now compare all methods on public benchmarks using three metrics: F1-score, ROC-AUC, and Precision-Recall AUC. Figure 3B has also been updated accordingly. In the antibody screening section, we have introduced new metrics based on the 9mer screening.

3. Data Availability and Reproducibility:

To ensure transparency and reproducibility, it is important to make the curated dataset and full analysis code publicly available. This would allow other researchers to validate, ensuring the model's results are robust.

The GitHub Repo is now publicly available (<https://github.com/pfizer-opensource/HLAIIIPred>) and includes python scripts, model weights, and the path to the curated data. The training data is available at: <https://zenodo.org/records/15299217>.

4. Peptide Length Considerations:

Most importantly, while the model's ability to include peptides with varying lengths, there are concerns about potential bias towards shorter peptides that might artificially inflate performance. The manuscript should provide more detailed data on the distribution of peptides analyzed across different length categories. In particular, the selection of 9-mer peptides for MHC Class II binding should be well-justified, as most literature suggests that MHC Class II typically accommodates longer peptides (i.e., peptides longer than 10 amino acids). A stratified performance comparison based on peptide length would help address this concern.

We added two plots to the SI to show the performance of HLAIIIPred across different peptide lengths in the test set. In addition, Figure 5B highlights the performance of all models with respect to the peptide length.

It is important to note that the HLAIIIPred supports a broad peptide length range of 8-26 AAs. The selection of 9mer binding core is independent of the peptide length and is well supported by the literature, specifically for the AAs interacting with MHCII in the binding groove. This prior knowledge is incorporated into the training process to improve the model's explainability (e.g., as illustrated in the Figure 6B). We added the following sentence to the Introduction to clarify this point:

“Although the length of peptides presented by HLAII can vary from 8 to 26 AAs, the majority of the binding occurs in a contiguous segment of 9 AAs, which is referred to as the peptide binding core¹⁸.”

This is followed by the sentence:

“We seek to identify binding cores along with peptide presentation in order to: ...”

Minor Comments:

1. Typos:

There are typos throughout the manuscript, such as “DQA101:01DQB105-02” and “50This”. Please carefully check.

Thank you – we have corrected all typos.

2. Inconsistent Information:

In the results section, the manuscript states that the true positive rate (TPR) of HLAIPred for all 42 alleles in the test set is greater than 50%. However, this is inconsistent with the data presented in Figure 3A, where not all alleles with a TPR above 0.5. Additionally, the abstract mentions a “16% increase in prediction of presented peptides compared to the second-best model on a set of unseen peptides presented by less frequent alleles,” but I could not find the corresponding information in the main text. This discrepancy should be addressed to ensure consistency.

The greater than 50% TPR for all 42 alleles, except DQA1*01:01-DQB1*05-02, was noted in the previous version of the manuscript. We have now updated the text based on the new results (and in response to Reviewer #2). The 16% improvement claim remains valid. We have now added the following to the Results section to clearly highlight this.

“As shown in Figure 3A(top), HLAIPred predicts presented peptides with greater accuracy compared to the previous models across 31 monoallelic samples. As shown in Figure 3A (bottom), 10 alleles (highlighted in red) are highly underrepresented in the training dataset, with less than 1% of total size. Among those 10 alleles, HLAIPred outperforms all other models in predicting pHLAIs for 8 of them, while Graph-pMHC is better than HLAIPred for only 2 alleles. Notably, the true positive rate of HLAIPred for all 42 alleles is 69%, while it is 58% for Graph-pMHC, 54% for MixMHC2pred-2.0, and 35% for NetMHCIIpan-4.1. This demonstrates an increase of at least 16% in the prediction of presented peptides.”

3. Definition Clarity:

In the reference section, specifically in the data for ADA Immunogenic, it would be helpful to define the term "pair" more clearly to improve readability. Additionally, it would be helpful to clarify how possible antibodies with identical sequences can exhibit different ADA responses. This clarification would help readers better understand the context of the results.

We revised the sentence in the Methods section to state:

“We identify 5 pairs of antibodies (i.e., 10 data points in total) with identical sequences but different ADA responses.”

Each pair consists of two antibodies with identical sequences but different ADA responses. We also added the following sentence to comment on the potential source of this data difference:

“These data differences are most likely due to different clinical conditions, such as patient-specific immune variability, if not simply due to inherent experimental or data collection errors.”

Conclusion:

While HLAIIIPred presents promising advancements in predicting peptide-HLA interactions, addressing these comments will significantly enhance your work's clarity, accuracy, and impact.

We sincerely appreciate your thorough review and valuable suggestions to improve the manuscript.

Reviewer #2 (Remarks to the Author):

The authors present a peptide-MHCII sequence-based binding predictor exploiting transformers and cross-attention. They compare their model against state-of-the-art predictors on peptide presentation and immunogenicity assessment over four experiments. The work includes a well-needed data-focused approach and attention to the out-of-distribution performance issues. The manuscript is well-written, the figures are good quality and the provided code is clean and easily understandable.

We appreciate your comments; they have greatly contributed to improving the quality and clarity of our manuscript.

My concerns are mainly focused on reproducibility and AI explainability.

Major concerns:

1. In the monoallelic sample comparison experiment, while mentioned by the authors, the actual overlap between train and test set alleles distribution is not shown. To properly understand the generalization capability of HLAIIIPred (and compare it with the other mentioned predictors), it is important to know the actual distance of the test set from the

training set. I suggest the authors add a figure (supplementary would be fine) showing the abundance of each allele from the dataset they curated from Strazar et al. compared with the same alleles abundance in their training set, and (only if possible) also the training sets used by the other software the authors compared with. I suggest the authors use the alleles pseudosequences when checking which alleles are "missing" from the training set, instead of simple allele type names that might change for mutations in non-binding-crucial parts of the MHC sequence. Pseudosequences can also be used to calculate the average or minimum distance of each "missing" allele from the training set. In this way, a reader can understand how much the training set(s) and architecture(s) contribute to the performances shown in Fig 3A.

Thank you for this excellent suggestion! We have added dataset size information to Figure 3A, along with the corresponding explanation in the Results section,

“As shown in Figure 3A(top), HLAIPred predicts presented peptides with greater accuracy compared to the previous models across 31 monoallelic samples. As shown in Figure 3A (bottom), 10 alleles (highlighted in red) are highly underrepresented in the training dataset, with less than 1% of total size. Among those 10 alleles, HLAIPred outperforms all other models in predicting pHLAIs for 8 of them, while Graph-pMHC is better than HLAIPred for only 2 alleles. Notably, the true positive rate of HLAIPred for all 42 alleles is 69%, while it is 58% for Graph-pMHC, 54% for MixMHC2pred-2.0, and 35% for NetMHCIIpan-4.1. This demonstrates an increase of at least 16% in the prediction of presented peptides.”

As suggested, we use pseudo-sequences to count the number of unique alleles. In addition, we not only removed exact matches but also ensured no 9mer overlap between epitopes from the Strazar dataset and the training data, recycling the removed data back into training. Consequently, we no longer observe alleles entirely absent from the training data. Instead, we considered a threshold of less than 1% frequency to identify underrepresented alleles.

2. I highly recommend the authors release the final data they used in this work. While it is great to have the code available on GitHub, parsing codes can have issues and might no longer work when a source database is updated. For this reason, releasing csv/tsv/xlsx files containing the training and test datasets (from public sources) used in this work is important for reproducibility and results interpretation.

The GitHub Repo is now publicly available (<https://github.com/pfizer-opensource/HLAIPred>) and includes python scripts, model weights, and the path to the curated data. The training data is available at: <https://zenodo.org/records/15299217>.

3. In the discussion, the authors suggest that HLAIPred is capable of learning

intermolecular interactions for single AAs/positions as opposed to fully-connected models which only rely on abstract combinations of features. Is it possible then for the authors to plot the attention weights of HLAIIIPred, maybe in a supplementary figure, to support this point? If HLAIIIPred is learning biologically significant interactions, this should be reflected in the attention weights. These should show strong attention between, for instance, the anchor residues of a peptide and the pocket-forming residues they interact with on the MHC side.

We have added Figure 3C and 3D to address this comment. Here is the corresponding information in the main text,

“In order to investigate the HLAIIIPred design choices, we examined the agreement between HLAIIIPred cross-attention and contact maps for 50 pHLAII complexes (see Figure 3 C and D). Figure 3D illustrates the percentage of contact maps recovered by top-3 HLAIIIPred attention ranges from 20% to 50% for alpha chains and 10% to 40% for beta chains. Given that the models are trained solely on peptide and allele sequences, these numbers show that the attention heads are able to recover a significant portion of the structural contact maps.”

Minor concerns:

1. In the first section of the results, the authors mention that weights from peptide clusters are used to modify the loss function. This sounds like a great idea in theory, but I wonder how much it is actually affecting the results. From the model training history provided, it's noticeable that in the first epoch of training, HLAIIIPred already shows a remarkably low loss of 0.1, given the task.

I think this work could largely benefit from training HLAIIIPred using only the BCE loss (i.e. without multiplying it with the peptide clusters weights) and showing/reporting the difference in initial loss and final model performances. If the loss should significantly increase upon removing the clusters' weights, the authors would have a stronger point on how their data-centred approach resulted in a better predictor.

I understand training HLAIIIPred takes a long time, so I leave this as optional to the authors.

We added Figure 3D to compare the performance of HLAIIIPred models trained with and without cluster weights and summarized the findings in the main text:

“In addition, we analyzed the effect of cluster weights on the model performance by training two versions of the model: with and without incorporating cluster weights during training. Figure 3E shows that the ROC-AUC of test dataset is evenly distributed across different cluster weights when cluster weights are applied in the loss function. Conversely,

the ROC-AUC favors larger clusters (i.e., subset of data with smaller cluster weights) when cluster weights are not considered during training.”

Please note that the loss value mentioned in your comment is after multiplication by cluster weights (i.e., they represent the customized BCE loss), which explains why the values appear notably low.

2. In the Introduction "Polyallelic nature of the data", why would the availability of more monoallelic data magnify the challenge of negative samples? Monoallelic data will provide just as many negative samples as multiallelic data (none). Do the authors mean here that having more MS data (independently of these being mono- or multi-allelic) increases the imbalance between positives and negatives, simply because there are no negatives? If so, I suggest rephrasing this sentence, as now it seems to imply that monoallelic samples specifically provide less negative data than multiallelic ones do.

Thanks for pointing out this statement – we agree that this may be interpreted subjectively. What we intended to convey is basically due to the uncertainty associated with the selection of negative data. Otherwise, we don't introduce any data imbalance in terms of the ratio of negatives to positives. The reason behind this uncertainty is that with the polyallelic data a random decoy can be screened against multiple alleles (one-to-many mapping), and with proper regularization, any error may be tolerated easier. In contrast, monoallelic data involves a one-to-one mapping, making any error in decoy selection to be implied more deterministic than probabilistic. We understand that this may come across as a subjective discussion, primarily focused on our training regularization. So, we decided to remove this argument from the main text.

Typos and miscellaneous:

1. Typo: Introduction, first paragraph, "from a genomic perspective"

We corrected the typo.

2. In the methods, when describing HLA-II pseudosequences, NetMHCIIpan-3.0 is missing the reference.

We reintroduced the reference.

Reviewers' comments:

Please note that Reviewer #1 has not raised any new concerns.

Reviewer #2 (Remarks to the Author):

The authors extensively replied to all my comments and questions, and provided very interesting additional results. The publication of train, validation and test data allows for better reproducibility and analysis of this work. I just have one more comment, but I trust the authors can adjust the text without the need for a new round of reviews. Moreover, I encountered a small issue with the github/zenodo links, explained below.

Besides this, I have no further comments, and I wish the authors a swift path to publication with this very nice work.

We sincerely appreciate your thorough review and valuable suggestions to improve the manuscript.

Major comment:

In the last section of the results, "HLAIIPred can prioritize vaccine neoantigens and predict binding specificities of HLAII alleles.", the authors defend the applicability of sequence-based approaches over structure-based approaches with multiple arguments, including "the poor performance of structure prediction models in terms of distinguishing between the presented and not-presented peptides". I do not think this conclusion is fair to the article the authors cite, which achieved almost exactly netmhcpan's performances (1% difference) by adding a small predictor on top of alphafold. I would not refer to such performances as "poor", especially considering they rely on a general-purpose modeling tool like Alphafold, which can be easily adapted to minimize the errors it makes when predicting pMHC structures. Moreover, structure-based methods have shown larger generalization power than sequence-based ones (<https://www.nature.com/articles/s42003-024-07292-1>). Similarly, having limited structural templates only poses an issue when trying to predict binding cores because it limits the benchmark set we can use for testing, but the available templates are more than enough to generate large libraries of good 3D models (showcased by softwares like PANDORA or APE-Gen2.0). I would rather encourage the authors to focus their point on the speed and thus large-scale applicability of sequence-based methods, which remain vital when screening large patients mutanomes. I can see the performance gap being filled very soon, quickly rendering such point moot, but it will be far more challenging for structure-based methods to achieve similar speeds than sequence-based ones. The authors could strengthen this point by adding a mention to the time taken by each tool on one pMHC-II case: ~10 minutes in tfold (on a A100 GPU) and probably less than a second for HLAIIPred.

Thank you for highlighting this important point. We also discuss this point in the Discussion section:

"However, these models are not able to distinguish between the presented and not-presented peptides, unless they are directly fine-tuned on the immunopeptidomics data⁴³. Considering the classification accuracy of such models and their computational cost, we believe the sequence-based approaches are still more practical for high-throughput screening of proteins and

antibodies. However, the structure-based models can be used as a complementary tool to the sequence-based models to improve the *generalizability*⁴⁴ and prediction of the binding core residues. ...”

In the Results section (subsection: HLAIIIPred can prioritize vaccine neoantigens and predict binding specificities of HLAII alleles.), the comment regarding the poor performance was intended to focus on structure prediction models without any further finetuning on the downstream task (i.e., MHC presentation prediction). Since a brief discussion at the end of the results paragraph is not helpful here, we decided to drop the comment on the performance and only provide a proper discussion in the Discussion section.

The modified sentence in the Results section looks like this:

We argue that the application of sequence-based pHLAII models remains justifiable, taking into account the computational cost of the TFold pipeline, which takes minutes to run compared to milliseconds for HLAIIIPred on the same GPU device.

Minor:

The links to github and zenodo do not work properly. The github link should be "<https://github.com/pfizer-opensource/HLAIIIPred>" but the one in the manuscript now is "<https://github.com/pfizeropensource/HLAIIIPred>" without a dash, which returns 404. Similarly, if the zenodo link is clicked on instead of manually copy-pasted, it includes the fullstop at the end ("[https://zenodo.org/records/15299217.](https://zenodo.org/records/15299217)") which directs to an empty page. Just adding a space between link and fullstop will solve this issue.

We have addressed both issues. The dash in the first link is currently positioned at the end of the line in our LaTeX version, which might be mistaken for an end-of-line dash. We will ensure that this issue does not persist in the final publication format.